# THE END OF MANUAL DECODING: TOWARDS TRULY END-TO-END LANGUAGE MODELS

**Zhichao Wang**[1,2][*]    **Dongyang Ma**[2][*]    **Xinting Huang**[2]    **Deng Cai**[2]    **Tian Lan**[2]

**Jiahao Xu**[2]    **Haitao Mi**[2]    **Xiaoying Tang**[1,3,4][†]    **Yan Wang**[2][*]

[1]School of Science and Engineering, The Chinese University of Hong Kong (Shenzhen)

[2]Tencent AILab

[3]Shenzhen Future Network of Intelligence Institute (FNii-Shenzhen), Shenzhen, China

[4]Guangdong Provincial Key Laboratory of Future Networks of Intelligence, Shenzhen, China

zhichaowang@link.cuhk.edu.cn   dongyangma@tencent.com
tangxiaoying@cuhk.edu.cn   yanwang.branden@gmail.com

## ABSTRACT

The "end-to-end" label for LLMs is a misnomer. In practice, they depend on a non-differentiable decoding process that requires laborious, hand-tuning of hyperparameters like temperature and top-p. This paper introduces *AutoDeco*, a novel architecture that enables truly "end-to-end" generation by learning to control its own decoding strategy. We augment the standard transformer with lightweight heads that, at each step, dynamically predict context-specific temperature and top-p values alongside the next-token logits. This approach transforms decoding into a parametric, token-level process, allowing the model to self-regulate its sampling strategy within a single forward pass.

Through extensive experiments on eight benchmarks, we demonstrate that *AutoDeco* not only significantly outperforms common decoding strategies but also achieves performance comparable to an oracle-tuned baseline derived from "hacking the test set"—a practical upper bound for any static method. Besides, we demonstrate an emergent capability for instruction-based decoding control: the model learns to interpret natural language commands (e.g., "generate with low randomness") and adjusts its predicted temperature and top-p on a token-by-token basis, which may open a new paradigm for steerable and interactive LLM decoding. Our model and code are available at: `https://github.com/Zacks917/AutoDeco`

## 1 INTRODUCTION

LLMs have become the de-facto standard in NLP, yet the quality of their generated text hinges on a surprisingly manual and heuristic process: the selection of decoding hyperparameters. Parameters like temperature, top-p, and top-k must be carefully chosen through a task-dependent process of manual sweeps and post-hoc filtering (Shi et al., 2024). This not only incurs significant computational and human costs but also profoundly impacts the final output's creativity, diversity, and factual correctness, undermining the promise of a truly "end-to-end" system.

This reliance on static, hand-tuned parameters creates fundamental bottlenecks. Firstly, the search for an optimal configuration is widely acknowledged as a laborious process because the ideal settings are highly task-dependent; commercial API providers like DeepSeek, for instance, explicitly recommend different temperature settings for distinct application scenarios[1]. However, this problem, runs even deeper: a single static configuration is inherently suboptimal because the ideal level of stochasticity varies dramatically within a single generation. For instance, a model might need high creativity to

---

[*] Equal Contribution

[†] Corresponding Author

[1]https://api-docs.deepseek.com/quick_start/parameter_settings

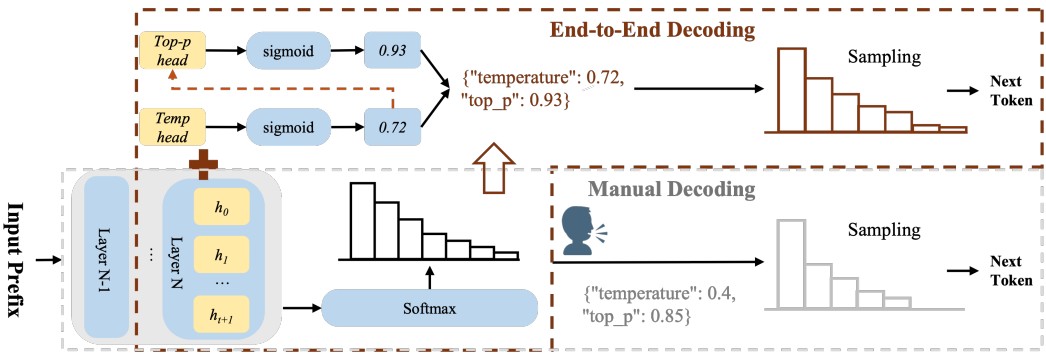

Figure 1: An overview of our proposed end-to-end decoding architecture compared to manual decoding. Our method dynamically predicts temperature and top-p values from the model's hidden states for each generation step. In contrast, manual decoding (bottom) relies on a single set of static, predefined hyperparameters for the entire sequence generation.

explore initial reasoning paths but high precision to deliver the final answer. This on-the-fly control is, by design, impossible for current LLMs to achieve natively. Consequently, the prevailing static decoding paradigm is a solution as inefficient as it is ineffective, forcing a one-size-fits-all strategy onto a problem that demands dynamic adaptation.

In this paper, we propose *AutoDeco*, a novel architecture that creates a truly "end-to-end" language model capable of controlling its own decoding process. As illustrated in Figure 1, we augment the standard transformer with lightweight, dedicated prediction heads. At each decoding step, these *AutoDeco* heads leverage the model's current hidden state to dynamically predict the optimal sampling parameters for the next token. This seamlessly integrates hyperparameter selection into the model's forward pass, creating a self-regulating inference pipeline that adds nearly-zero latency.

We validate our approach by integrating *AutoDeco* into major model families, including Qwen, Llama, GPT, and DeepSeek, requiring only a brief fine-tuning process of 400 steps. Across eight distinct benchmarks, the results are striking: *AutoDeco* not only consistently outperforms standard default decoding settings but also matches or surpasses the performance of meticulously expert-guided tuning (an oracle-tuned baseline derived from "hacking the test set") hyperparameters. An important secondary benefit of our architecture is the observed capacity for instruction-based decoding control, which is learned unexpectedly during the "end-to-end" optimization. When prompted with a meta-instruction like, "Please ensure that the diversity of your output is low," the model immediately responded by lowering its average predicted temperature and top-p values by 0.11 and 0.09, respectively. This demonstrates that *AutoDeco* does not merely automate a tedious process; it endows the model with a new, intuitive way to interpret and act on user intent.

Our contributions are four-fold: **(i)** We propose *AutoDeco*, a novel and lightweight architecture, along with an efficient strategy to train its prediction heads, that makes LLM generation truly "end-to-end" by dynamically predicting decoding parameters at each step. **(ii)** We demonstrate through extensive experiments that *AutoDeco* consistently matches or exceeds the performance of expert-guided tuning, static hyperparameters across eight benchmarks and multiple model families. **(iii)** We demonstrate for the first time that an LLM's decoding can be controlled by natural language. **(iv)** We release a set of *AutoDeco* heads, trained on the most widely adopted open-source models, providing the community with a streamlined, drop-in solution for immediate deployment.

## 2 AUTODECO

The foregoing discussion raises two fundamental questions that frame the core inquiry of this work:

First, how can we train the *AutoDeco* heads without any token-level "ground-truth" labels for the optimal temperature and top-p values? Second, how can these predictions be integrated into inference without adding computational latency? This section details our solutions to both.

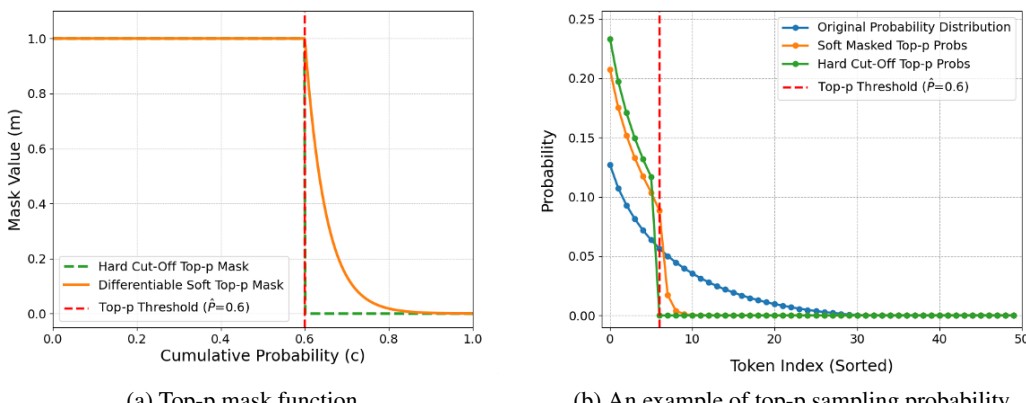

(a) Top-p mask function.       (b) An example of top-p sampling probability.

Figure 2: Comparison of the differentiable "soft" top-p sampling (decay steepness $\alpha = 30$) with the standard hard-cutoff method. (a) illustrates the standard hard-cutoff mask, which has a non-differentiable step, against our proposed smooth and differentiable "soft" mask. (b) shows the effect of applying both masks to an example original probability distribution, where the "soft" mask method produces a differentiable probability distribution suitable for "end-to-end" training.

In Section 2.1, we will introduce our training strategy and explain how we train both heads in an "end-to-end" manner. Then, in Section 2.2, we will walk through our inference process. The *AutoDeco* modifies the model's final output probabilities internally—a design that adds absolutely no extra latency. The result is a model that can be used almost exactly like a standard one, requiring only a "1-line-change" in a user's code to unlock its dynamic decoding capabilities.

## 2.1 TRAINING STRATEGY

The central challenge in training *AutoDeco* is the absence of token-level "ground-truth" labels for sampling parameters. A natural approach would be to optimize the temperature and top-p heads directly from the final cross-entropy loss of the generated tokens. However, this path is obstructed by the standard top-p sampling algorithm. Its "hard cutoff"—retaining only the smallest set of tokens whose cumulative probability exceeds a threshold—is a non-differentiable operation, which severs the gradient flow from the loss back to the top-p head.

To overcome this, we introduce a novel, differentiable "soft" top-p mechanism that is used during training, enabling a fully "end-to-end" optimization strategy. Traditional top-p sampling methods assign a probability of zero to all tokens beyond the top-p threshold, while our approach is different: for tokens that fall outside the top-p threshold, we apply a differentiable weight scaling. The further a token is from the threshold, the more its probability is scaled down, eventually approaching zero.

The following is the training data stream:

1. **Temperature-Scaled Probabilities:** First, we scale the predicted logits $\mathbf{l}$ to compute the initial probability distribution $\mathbf{p}$ using the predicted temperature $\hat{T}$.

$$\mathbf{p} = \text{softmax}\left(\frac{\mathbf{l}}{\hat{T}}\right). \tag{1}$$

2. **Differentiable Mask Generation:** After sorting the probabilities $\mathbf{p}$ and calculating their cumulative sum $\mathbf{c}$, we generate a "soft mask" $\mathbf{m}^{(\text{sorted})}$. This is done in a single step that combines the thresholding and decay logic:

$$\mathbf{m}^{(\text{sorted})} = \exp\left(-\alpha \cdot \text{ReLU}(\mathbf{c} - \hat{P})\right), \tag{2}$$

Here, $\alpha$ is a hyperparameter that controls the steepness of decay. As shown in Figure 2a, this formulation ensures that for tokens inside the nucleus (where $\mathbf{c} < \hat{P}$), the ReLU term is zero, resulting in a mask value of 1. For tokens outside, the mask value smoothly decays towards zero as their cumulative probability further exceeds $\hat{P}$.

3. **Final Probability Distribution:** The "soft mask" $\mathbf{m}$ (unsorted to match the original vocabulary order) is applied to the initial probabilities, and the result is re-normalized to form the final, differentiable distribution $\tilde{\mathbf{p}}$:

$$\tilde{\mathbf{p}} = \frac{\mathbf{p} \odot \mathbf{m}}{\sum(\mathbf{p} \odot \mathbf{m}) + \epsilon}, \tag{3}$$

where $\epsilon$ is a small constant for numerical stability. In Figure 2b, we provide an example with a vocabulary size of 50 to illustrate how the model's predicted probability distribution changes after the application of our "soft" top-p sampling. As the probability of the token exceeding $\hat{P}$ decreases gradually and differentially, the "soft" top-p sampling becomes the final piece of the puzzle for the *AutoDeco*'s "end-to-end" training.

**Training.** As the entire process is differentiable, gradients from the standard cross-entropy loss are backpropagated to simultaneously update the parameters of both the temperature and top-p heads, allowing the model to learn its own optimal, context-specific decoding strategy by directly optimizing for the final task objective.

Theoretically, these two heads could be trained from the pre-training stage. However, in this paper, we build upon a pre-trained LLM, freezing its base parameters and solely training the *AutoDeco* heads. While training these heads on SFT data provides a strong baseline, we find that applying some certain de-biasing operations to the data can further enhance model performance and robustness:

- **Easy-Token Masking.** For many tokens, the base model's greedy prediction already matches the ground-truth. These "easy" tokens often yield an optimal temperature $\hat{T}_t^*$ near zero, biasing the head to be overly conservative. To mitigate this, we randomly mask the training loss for a large fraction (e.g., 60%) of these positions, forcing the model to learn from more challenging and informative examples.

- **Dynamic Fine-Tuning.** Conversely, a naive fine-tuning approach can cause the temperature head to predict unexpected large values for uncertain tokens. We incorporate Dynamic Fine-Tuning (Wu et al., 2025), which re-weights the training loss to focus on tokens where the model has a reasonable prior. This teaches the head to apply high temperatures more judiciously in situations of calibrated uncertainty, rather than being skewed by outlier signals.

## 2.2 INFERENCE: DYNAMIC DECODING

At the heart of *AutoDeco* lies a design optimized for efficiency. By seamlessly integrating all dynamic adjustments into the model's standard forward pass, it avoids any separate, costly computational steps. This architecture results in a negligible latency overhead, typically adding only 1-2% to the total generation time. As illustrated in Figure 1, the process for each token generation step is as follows:

1. **Compute Hidden State:** The base LLM computes the final hidden state $\mathbf{h}_t$.

2. **Predict Decoding Parameters:** In parallel, the standard `lm_head` computes the logits while the *AutoDeco* heads predict the dynamic parameters. The temperature is predicted directly from the hidden state. Crucially, the top-p head then uses both the hidden state *and* the just-predicted temperature as input:

$$\hat{T}_t = temp\_head(\mathbf{h}_t), \quad \hat{P}_t = top\text{-}p\_head(\mathbf{h}_t, \hat{T}_t). \tag{4}$$

This micro-dependency, shown as a dashed arrow in Figure 1, allows for a more nuanced interplay between the two parameters.

3. **Internal Probability Modification:** The model immediately uses the predicted $\hat{T}_t$ and $\hat{P}_t$ to internally rescale and filter the logits, producing a final, dynamically-adjusted distribution.

**Latency and Simplicity.** The *AutoDeco* heads (simple 2-layer MLPs) add negligible computational overhead compared to the massive transformer layers. This internal architecture results in only 1-2% additional latency and makes usage incredibly simple, and ensures seamless integration, allowing an *AutoDeco*-enabled model to serve as a drop-in replacement for its standard counterpart, requiring no modifications to the user's existing generation logic.

# 3 EXPERIMENTS

We conduct extensive experiments to validate *AutoDeco*, structuring our evaluation around its core contributions to performance, efficiency, and a surprising capability that emerged as a byproduct.

- In Section 3.2.1, we demonstrate the superior performance of *AutoDeco*. It not only substantially outperforms standard, non-expert decoding baselines (Greedy Search and Default Sampling) but also matches or even slightly surpasses the performance of optimal static hyperparameters found through an exhaustive expert-guided tuning.

- Following this, in Section 3.2.2, we analyze its practical efficiency and confirm that *AutoDeco* introduces a minimal computational burden, with a marginal latency increase of 1-2% and a negligible memory footprint.

- A noteworthy finding is presented in Section 3.3: the emergent capability of AutoDeco to interpret natural language commands to dynamically steer its own generation style. This development is a significant step towards more intuitive and controllable AI.

## 3.1 EXPERIMENTAL SETUP

**Models.** To demonstrate broad applicability, we select representative models from three of the most popular open-source model families. All *AutoDeco* heads are trained on top of the official pre-trained checkpoints of these models:

- **Llama-3.1-Nemotron-Nano-8B-v1**(Bercovich et al., 2025): A general-purpose model from the widely-used Llama family, developed by Nvidia (hereinafter Llama-Nemotron-8B).

- **R1-Distill-Qwen-7B**(Guo et al., 2025): A distilled model from the Qwen family developed by DeepSeek, known for its strong reasoning capabilities.

- **Qwen3-30B-A3B-Instruct-2507**(QwenTeam, 2025): An advanced MoE architecture instruct (non-thinking) model from Qwen. (hereinafter Qwen3-30B-Instruct)

- **Qwen3-235B-A22B-Thinking-2507**(QwenTeam, 2025): An advanced MoE architecture Thinking model from Qwen. (hereinafter Qwen3-235B-Thinking)

- **OpenAI-GPT-OSS-20B**(Agarwal et al., 2025): A MoE model with 20B parameters released by OpenAI. The reasoning effort is set to medium by default.

More models, including **DeepSeek-V3.1-Terminus (with multi-token prediction)** (DeepSeek-AI, 2024), and results can be found in the Appendix 9.

**Datasets.** The models are trained on a focused domain and evaluated on a wide range of tasks to test for generalization.

- **Training Data:** The *AutoDeco* heads are trained on a specialized dataset of reject sampling trajectories. These trajectories were generated by sampling solutions from all the base models on problems from the DeepMath-103K dataset (He et al., 2025).

- **Evaluation Benchmarks:** We evaluate on a diverse suite of eight benchmarks, split into two categories to assess both in-domain and out-of-domain performance:
  - **In-Domain (Math):** AIME (24+25), BRUMO25, HMMT25 (Balunović et al., 2025), and BeyondAIME (ByteDance-Seed, 2025).[2]
  - **Out-of-Domain (General Tasks):** GPQA-Diamond (Rein et al., 2024) and MMLU-Pro (Wang et al., 2024b) (QA) , LiveCodeBenchV6 (Jain et al., 2024) (Code), and IFEval (Zhou et al., 2023) (Instruction Following).

**Baselines and Evaluation.** We evaluate *AutoDeco* against two standard, non-expert decoding strategies: **Greedy Search** and **Default Sampling** ($\hat{T} = 1.0, \hat{P} = 1.0$). Furthermore, to establish a practical upper bound, we also compare against an **Expert-Guided Tuning**. It is crucial to note that

---

[2]We focus on these recent, hard benchmarks to mitigate the risk of data leakage issues in older datasets.

Table 1: Pass@1 accuracy on mathematical reasoning benchmarks. *AutoDeco* consistently outperforms both Greedy Search and Default Sampling methods across various models.

| Model | Method | AIME | BRUMO25 | HMMT25 | BeyondAIME | Average |
|---|---|---|---|---|---|---|
| Llama-Nemotron-8B | Greedy Search | 55.00 | 60.00 | 26.67 | **37.00** | 44.67 |
| | Default Sampling | 51.24±1.31 | 60.68±1.21 | 30.89±1.29 | 32.65±0.51 | 43.86 |
| | *AutoDeco* (Ours) | **56.07±1.12** | **63.23±1.04** | **34.37±1.33** | 34.51±0.54 | **47.05** |
| R1-Distill-Qwen-7B | Greedy Search | 40.00 | 43.33 | 20.00 | 24.00 | 31.83 |
| | Default Sampling | 43.62±1.06 | 51.38±1.08 | 22.50±0.88 | 24.87±0.72 | 35.59 |
| | *AutoDeco* (Ours) | **47.57±1.20** | **53.93±1.31** | **24.46±0.76** | **26.98±0.75** | **38.24** |
| Qwen3-30B-Instruct | Greedy Search | 65.00 | 70.00 | 36.67 | 45.00 | 54.17 |
| | Default Sampling | 66.95±1.44 | 70.99±1.11 | **44.69±0.81** | 47.02±0.33 | 57.41 |
| | *AutoDeco* (Ours) | **68.22±0.85** | **71.36±0.83** | 43.15±1.10 | **47.15±0.20** | **57.47** |
| Qwen3-235B-Thinking | Greedy Search | 80.00 | 83.33 | 63.33 | **53.00** | 69.92 |
| | Default Sampling | 80.76±0.43 | 82.40±0.69 | 63.41±1.20 | 50.05±0.56 | 69.16 |
| | *AutoDeco* (Ours) | **82.79±0.72** | **84.14±1.27** | **66.07±1.10** | 51.88±0.62 | **71.22** |
| OpenAI-GPT-OSS-20B | Greedy Search | 56.67 | 73.33 | **56.67** | 37.00 | 55.92 |
| | Default Sampling | 69.65±1.33 | **75.50±1.11** | 50.91±2.76 | 47.11±0.23 | 60.79 |
| | *AutoDeco* (Ours) | **72.72±1.22** | 75.21±0.83 | 53.00±1.02 | **47.32±0.37** | **62.06** |

Table 2: Pass@1 accuracy on general-domain benchmarks. *AutoDeco* shows exciting generalization performance across General QA, Code Generation, and Instruction Following tasks.

| Model | Method | GPQA-Diamond | MMLU-Pro | LiveCodeBenchV6 | IFEval | Average |
|---|---|---|---|---|---|---|
| Llama-Nemotron-8B | Greedy Search | **51.01** | 52.00 | 19.17 | **71.53** | 48.43 |
| | Default Sampling | 44.93 | 54.00 | 21.22 | 65.25 | 46.35 |
| | *AutoDeco* (Ours) | 50.52 | **55.64** | **21.68** | 71.02 | **49.72** |
| R1-Distill-Qwen-7B | Greedy Search | 37.87 | 47.20 | 14.98 | 32.90 | 33.24 |
| | Default Sampling | 47.41 | 47.65 | 15.40 | 32.35 | 35.70 |
| | *AutoDeco* (Ours) | **48.91** | **50.75** | **15.46** | **33.90** | **37.26** |
| Qwen3-30B-Instruct | Greedy Search | 65.86 | 78.00 | 47.75 | **83.73** | 68.84 |
| | Default Sampling | 69.82 | 76.25 | 48.52 | 81.52 | 69.03 |
| | *AutoDeco* (Ours) | **69.96** | **78.38** | **49.80** | 82.81 | **70.24** |
| Qwen3-235B-Thinking | Greedy Search | 77.78 | **81.00** | 77.25 | 31.98 | 67.00 |
| | Default Sampling | 80.81 | 80.33 | 77.47 | 31.61 | 67.56 |
| | *AutoDeco* (Ours) | **81.13** | 79.20 | **78.10** | **32.90** | **67.83** |
| OpenAI-GPT-OSS-20B | Greedy Search | 59.60 | 67.00 | 69.69 | 29.94 | 56.56 |
| | Default Sampling | 65.67 | 68.00 | 70.15 | 30.68 | 58.63 |
| | *AutoDeco* (Ours) | **66.48** | **69.12** | **71.25** | **30.84** | **59.42** |

this expert-tuned baseline is an *oracle* setting, as it involves finding the optimal static hyperparameters by tuning on the test set—a process that is infeasible in real-world applications.

Our primary metric is Pass@1 accuracy, estimated via oversampling with 128 samples per problem (with 8 random seeds, 16 samples per seed). The maximum generation length is set to 32768.

## 3.2 MAIN RESULTS

We present our main findings separately for mathematical reasoning and open-domain question answering to provide a clear and detailed view of *AutoDeco*'s performance across different domains.

### 3.2.1 PERFORMANCE

**In-Domain Performance.** As shown in Table 1 *AutoDeco* consistently demonstrates a performance boost compared to Greedy Search and Default Sampling. For instance, on Llama-Nemotron-8B, it achieves an average score of 47.05, a substantial average improvement of nearly 2.8 absolute points over Default Sampling and Greedy Search.

One may notice that the performance gain from *AutoDeco* is less pronounced on Qwen3-30B-A3B-Instruct-2507 compared to other models. This may stem from Qwen3-30B-A3B-Instruct-2507, as a non-thinking-model, produces answers that are significantly shorter than the other models.

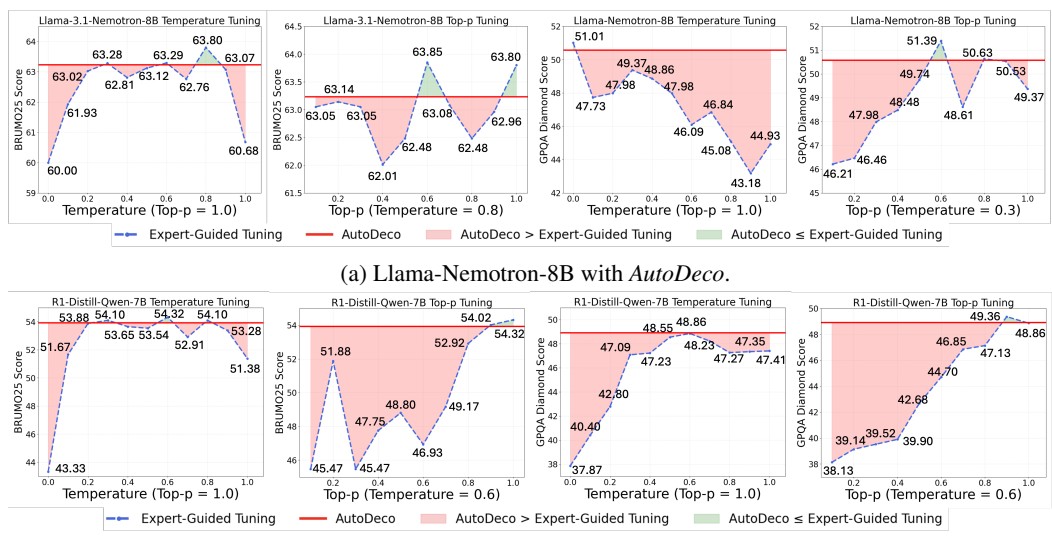

(a) Llama-Nemotron-8B with *AutoDeco*.

(b) R1-Distill-Qwen-7B with *AutoDeco*.

Figure 3: Expert-Guided Tuning Comparison with Search Interval of 0.1. Temperature is adjusted first (setting top-p to 1.0), and the selection is made based on the best performance of temperature to conduct the search for top-p. *AutoDeco* achieves competitive performance without requiring any prior empirical tuning or domain-specific expert knowledge.

Consequently, the sensitivity of task accuracy to variations in sampling parameters is substantially lower, a trend that is further demonstrated by the results in Table 2.

**Out-of-Domain Generalization.** More strikingly, despite being trained exclusively on mathematical reasoning, *AutoDeco* demonstrates powerful zero-shot generalization to a diverse set of out-of-domain tasks (Table 2). It consistently secures the highest average scores across general QA, code generation, and instruction following. This strong performance reveals two interesting patterns.

First, the magnitude of improvement is remarkably consistent across domains. For example, on R1-Distill-Qwen-7B, *AutoDeco* improves the average score on general tasks by 2.8 points over Default Sampling and Greedy Search—a gain even matching that seen in the math domain. This suggests that the benefits of dynamic decoding are fundamental and not tied to a specific task type.

Second, *AutoDeco* shows an ability to dynamically balance deterministic and stochastic strategies. On general tasks, Default Sampling is not always better than Greedy Search (e.g., on Llama-Nemotron-8B for GPQA-Diamond and IFEval). In these cases, *AutoDeco* learns to predict more deterministic, low-temperature parameters, allowing it to match or exceed the performance of the stronger greedy baseline. Conversely, when stochasticity is beneficial, it raises the temperature to outperform Default Sampling.

The above findings suggest that *AutoDeco* is not simply learning "what" to generate, but rather the fundamental "meta-skill of how" to generate text effectively. By training on a high-signal domain like mathematics, it learns universal principles for balancing exploration and exploitation. We also show that *AutoDeco* has the ability to adapt to different task requirements (Stability vs. Creativity) in Appendix 9. We will further discuss this in Sec. 3.3, and this finding challenges the conventional assumption that adaptive decoding requires broad, task-matched supervision, and instead points toward a more efficient, modular paradigm for real "end-to-end" controllable generation.

**Pass@$k$ Performance.** Some recent works (Yue et al., 2025; Chen et al., 2025) have highlighted a potential trade-off in the training of reasoning models, where achieving superb pass@1 accuracy can come at the expense of performance on pass@$k$ (for $k > 1$). To investigate this, we present an extended evaluation of our method on pass@$k$ ($k = 16, 32, 64$) accuracies.

With encouraging results, we find that the absolute improvements delivered by *AutoDeco* at higher $k$-values are consistent with, and at times even slightly greater than, those observed at pass@1. We show the numerical results in the Appendix 9.

It is important to note that for any given model, pass@$k$ accuracy is inherently much higher than pass@1 accuracy. Consequently, it is obvious that securing absolute performance gains becomes substantially more difficult, and a similar absolute improvement at pass@64 translates to a much larger relative error reduction, compared to pass@1. For example, on the OpenAI-GPT-OSS-20B model, we observe that the performance gains from *AutoDeco* are consistent across different $k$ values in pass@$k$ evaluations. More importantly, this consistent absolute gain translates to a significantly larger impact in higher-accuracy (when k is large) scenarios. The relative error reduction dramatically increases from **3.2%** at pass@1 to **30%** at pass@64. This demonstrates that as the task becomes easier for the baseline model (i.e., the error rate decreases at high $k$), the performance gains from our method become even more significant.

**Comparison with Expert-Guided Tuning.** In real-world applications, developers often undertake a laborious tuning process to find task-specific, optimal static hyperparameters. To assess how *AutoDeco* compares to this best-case scenario, we simulate an expert with an unfair advantage: access to a test-set oracle. As shown in Figure 3, we first perform a fine-grained search to find the optimal static temperature on the test set, and then, using that temperature, find the optimal top-p. This process represents the practical upper bound for any static decoding strategy.

The results are striking. *AutoDeco*'s single-pass performance is nearly identical to this oracle-tuned baseline, with the performance gap consistently less than one point across all models and datasets. Given that the Expert-Guided Tuning relies on "hacking the test set", a process impossible in any real-world scenario where the test data is unknown, we can confidently assert that *AutoDeco* is effectively superior to any feasible expert-tuning strategy in practice.

Furthermore, the figure highlights the fundamental limitation of static decoding: the optimal hyperparameters are extremely task-dependent. For instance, Llama-Nemotron-8B requires drastically different settings for BRUMO25 ($\hat{T} = 0.8, \hat{P} = 0.6$) versus GPQA-Diamond ($\hat{T} = 0.3, \hat{P} = 0.6$). However, in real-world scenarios, a model developer has no way to switch hyperparameters based on the user's query type. *AutoDeco* elegantly solves this problem. By achieving near-oracle performance automatically and on-the-fly for any task, it provides the optimal and, frankly, only practical solution for developers seeking robust, high-performance generation across diverse user inputs.

**Compatibility with Advanced Decoding Algorithms.** In practice, using only the model for next-token prediction might be inefficient. For industrial and production-level deployment and usage, speculative decoding is a very common and advanced technique, such as multi-token prediction (MTP) (DeepSeek-AI, 2024) that is widely used in the state-of-the-art models of various families (QwenTeam, 2025). Our *AutoDeco* is fully compatible with speculative decoding mechanisms:

- Theoretically, speculative decoding (such as MTP, which is currently applied in production-level models like DeepSeek-V3.1, Qwen3-Next) is fundamentally still of the form of autoregression. Sampling parameters such as Temperature and Top-p are widely and continuously employed within the speculative decoding framework to control the diversity and quality of the final accepted tokens.

- To empirically confirm this seamless integration, we conducted a MTP experiment on the advanced LLM DeepSeek-V3.1-Terminus. It shows that our method does not require any specific adaptation and will not break the process of speculative decoding. (Please refer to Appendix 9 for detailed results.)

**Ablation Study.** A natural question is what role the temperature and top-p heads play individually. To isolate their effects, we evaluate on AIME using R1-Distill-Qwen-7B to conduct an ablation study, with the results presented in Figure 4. The most striking finding is the remarkable effectiveness of each component in isolation. Using either the temperature head or the top-p head alone achieves an average performance gain of approximately 3-3.5 absolute points over the Default Sampling baseline.

Table 3: FLOPs, Memory Usage and latency (1k tokens) across various prompt length for R1-Distill-Qwen-7B with/without temp head and top-p head.

| Metrics | Method | 1k | 2k | 4k | 8k | 16k | 24k |
|---------|--------|-----|-----|-----|-----|------|------|
| FLOPs | Default Sampling | 2.89e+13 | 4.34e+13 | 7.23e+13 | 13.03e+13 | 24.61e+13 | 36.19e+13 |
| | *AutoDeco* (Ours) | 2.89e+13 | 4.34e+13 | 7.24e+13 | 13.03e+13 | 24.62e+13 | 36.20e+13 |
| Latency (s) | Default Sampling | 18.23 | 18.86 | 18.93 | 19.72 | 22.11 | 25.76 |
| | *AutoDeco* (Ours) | 18.84 | 19.10 | 19.43 | 20.03 | 22.36 | 26.05 |
| Memory (MB) | Default Sampling | 15546 | 16032 | 17130 | 19098 | 23182 | 27262 |
| | *AutoDeco* (Ours) | 15550 | 16036 | 17134 | 19102 | 23183 | 27266 |

This result is highly significant. It demonstrates that substantial improvements in decoding do not require a sophisticated architecture. A single, lightweight prediction head is sufficient to dramatically outperform standard static decoding methods.

Of course, while each head is powerful on its own, our results also confirm that the full *AutoDeco* model, with both heads, yields the best performance. They provide complementary benefits, allowing for even finer-grained control over the generation process to achieve optimal results.

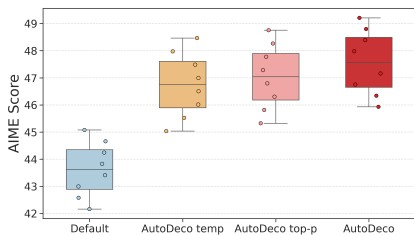

Figure 4: Ablation study on *AutoDeco* architecture designs. Joint optimization achieves the highest AIME Score.

### 3.2.2 EFFICIENCY

A critical advantage of *AutoDeco* is its computational efficiency. To quantify this, we evaluated its overhead against Default Sampling across three key metrics, with results summarized in Table 3.

The analysis shows that the additional computational burden is minimal. The FLOPs are virtually identical to the baseline, and the memory footprint increases by a mere 4 MB, an insignificant amount for modern hardware. The impact on latency is also negligible. This overhead remains consistently low regardless of prompt length, adding a consistent overhead of 0.29-0.6 s/k tokens, which translates to an average relative increase of just 1.7%.

These results empirically validate that *AutoDeco* is a lightweight enhancement. When considering the substantial performance gains and the convenience of automatic, task-agnostic hyperparameter tuning demonstrated in Sec. 3.2.1, this minor computational cost becomes trivial. *AutoDeco* thus presents a highly practical solution, offering significant benefits for a negligible price.

The analysis regarding training efficiency can be found in the Appendix 8.

### 3.3 EMERGENT CONTROL OF DECODING VIA NATURAL LANGUAGE

Beyond outperforming static methods, we observe that *AutoDeco* acquires a crucial capability: it learns to map abstract, high-level commands (such as instructions for diversity or certainty) directly to its internal decoding parameters. This learned instruction-based control enables the model to dynamically respond to user intent regarding the desired generation style, marking a significant step towards truly end-to-end and controllable generation. For this phenomenon, we conducted an in-depth evaluation and discussion, which are detailed in the Appendix 7.

## 4 RELATED WORKS

The process of generating text from a language model, known as decoding, is a critical step that significantly influences the quality of the output(Wang et al., 2025; Shi et al., 2024). One directly related work is Adaptive Decoding (Dhuliawala et al., 2024), which also introduces a learned decoding head. Their method focuses on predicting temperature and utilizes reinforcement learning (RL) for training. Our work differs in two main aspects: 1) we learn to control both temperature and top-p, and 2) we employ a fully differentiable pipeline that allows for direct "end-to-end" training from

the next-token prediction loss, instead of RL. The other existing decoding strategies can be broadly categorized into deterministic, stochastic sampling, and model-based approaches, most of which traditionally rely on static, predefined configurations.

**Deterministic Decoding.** Deterministic methods produce a single, reproducible output for a given input. The most fundamental of these is Greedy Search, which selects the token with the highest probability at each step. Another classic one is beam search, which maintains a "beam" of k most probable partial sequences to explore a larger search space (Sutskever et al., 2014; Graves, 2013). However, both of them are known to favor dull, high-frequency phrases (Vijayakumar et al., 2016), this results in their good performance on Machine Translation and QA tasks, but not suitable for open-ended generation tasks. A more recent line of deterministic methods, Contrastive Search(Su & Collier, 2022; Su et al., 2022), directly optimizes for open-ended generation quality by penalizing tokens that are too similar to previous tokens, effectively mitigating the degeneration problem.

**Stochastic Sampling.** To inject diversity, stochastic sampling methods are essential. These methods sample from the model's output probability distribution, which is typically modulated by some hyperparameters. However, unrestricted sampling can produce incoherent text. To counter this, truncation methods were developed. Top-K sampling(Fan et al., 2018) restricts the sampling pool to the $k$ most likely tokens, while the more adaptive Nucleus Sampling (top-p)(Holtzman et al.) selects the smallest set of tokens whose cumulative probability exceeds a threshold $p$. Despite their power, as our introduction highlights, finding the optimal configuration for these hyperparameters is a non-trivial, task-dependent manual process (Shi et al., 2024).

**Model-Based Decoding.** To gain more fine-grained control over generation, a third category of methods modifies the model's output distribution using external signals or auxiliary models. Early examples include **Plug-and-Play Language Models**, which leverage attribute models to steer generation towards desired topics (Dathathri et al.). More recently, **Contrastive Decoding** uses a smaller "amateur" model to steer a larger "expert" model away from generic text (Li et al., 2023; Chuang et al., 2023). Similarly, Speculative Decoding utilizes a faster "draft" model to generate sequences of tokens that are then verified by the larger model, significantly accelerating inference (Leviathan et al., 2023; Chen et al., 2023). There is also an art to verification methods (Liu et al., 2025). While they are effective, they still operate under a fixed algorithmic framework: the choice of the "guidance model" itself acts as another form of hyperparameter. For example, in contrastive decoding and speculative decoding, the authors suggest that using a smaller LM of the same architecture as the guidance model yields the best results.

Despite this rich landscape of research, a fundamental limitation persists: all these methods rely on a static decoding strategy. Whether it's a fixed algorithm (like Beam Search) or a fixed set of hyperparameters, this "one-size-fits-all" approach is inherently suboptimal. In contrast, *AutoDeco* proposes a paradigm shift. Instead of relying on fixed hyperparameters or predefined heuristics, we empower the model to dynamically control its own stochasticity at each generation step.

## 5 CONCLUSION AND FUTURE WORK

In this work, we challenged that the "end-to-end" label for LLM is a misnomer. We introduced *AutoDeco*, a truly "end-to-end" architecture that empowers models to dynamically control their own decoding strategy. By learning to predict token-level temperature and top-p values, *AutoDeco* transforms decoding from a manual, static process into a dynamic, self-regulating pipeline.

Our extensive experiments reveal three key contributions. First, *AutoDeco* demonstrates remarkable generalization, consistently outperforming standard decoding methods across diverse models and tasks, even matching oracle-tuned baselines without any task-specific tuning. Second, this performance is achieved with negligible computational overhead, making it a practical, drop-in enhancement for any transformer-based model. Additionally, we discovered an intriguing emergent capability: *AutoDeco* learns to interpret natural language commands to steer its own generation style, a foundational step towards more intuitive human-AI interaction.

Future Work. Our immediate future work involves jointly training the base model with *AutoDeco*. We believe this will address current limitations like imprecise prompt-based control and data biases—both likely consequences of a frozen backbone—thereby enabling more granular control over generation.

## ACKNOWLEDGMENTS

This work is supported in part by the Guangdong Basic and Applied Basic Research Foundation under Grant No. 2025A1515012968, in part by the Shenzhen Science and Technology Program under Grant No. JCYJ20240813113502004, in part by the National Natural Science Foundation of China under Grant No. 62001412, in part by Shenzhen Stability Science Program 2023, in part by the Guangdong Provincial Key Laboratory of Future Networks of Intelligence (Grant No. 2022B1212010001), and in part by the Shenzhen Key Lab of Crowd Intelligence Empowered Low-Carbon Energy Network (Grant No. ZDSYS20220606100601002).

## STATEMENTS

### ETHICS STATEMENT

The authors of this paper have read and agree to abide by the ICLR Code of Ethics. We believe that this work does not raise any significant ethical concerns. Our research did not involve experiments with human subjects, nor did it process sensitive personal data. All datasets used in our study are publicly available. We foresee no direct negative societal impacts from the methods and potential applications presented in this work.

### REPRODUCIBILITY STATEMENT

We are committed to ensuring the reproducibility of our research. We have provided comprehensive experimental details in the main paper and Appendix 6, including dataset preprocessing procedures, model architecture specifications, full training details, and all hyperparameter configurations. Furthermore, we will make our source code and model checkpoints publicly available.

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

CONTENTS OF THE PAPER

## 6    EXPERIMENTAL SETUP

**Training.**    For the training of all models with our AutoDeco framework, we employed a consistent hyperparameter configuration to ensure fair comparison. To efficiently manage memory and scale our experiments, we utilized the DeepSpeed library with the ZeRO Stage 3 optimization. The specific training settings are detailed below:

- **Training Framework:** DeepSpeed (ZeRO Stage 3) for DeepSeek-R1-Distill-Qwen-7B and Llama-3.1-Llama-Nemotron-8B-8B-Nano-v1. Megatron for the MoE model Qwen3-30B-A3B-Instruct-2507, Qwen3-235B-A22B-Thinking-2507, OpenAI-GPT-Oss-20B, OpenAI-GPT-Oss-120B, and DeepSeek-V3.1-Terminus.
- **Hardware:** 32 GPUs for DeepSeek-V3.1-Terminus, 8 GPUs for the others.
- **Batch Size:** A per-device batch size of 1 with 4 gradient accumulation steps, resulting in an effective global batch size of 32.
- **Optimizer:** AdamW
- **Learning Rate:** $5 \times 10^{-6}$.
- **Max Token Length:** 32768.
- **Decay Steepness $\alpha$:** 30.

For each task, we calculated the Pass@1 through oversampling (16 times). To ensure the results are solid, we do 8 runs on each experiment with different seeds. For all the tasks, our maximum generation length is 32768.

**Datasets.**    Our experimental configuration is detailed as follows:

- **MMLU-Pro:** We used a comprehensive and evenly distributed "lite" subset [3] for evaluation to ensure a balanced assessment across all subject areas.

---

[3]https://huggingface.co/datasets/koiwave/100MMLUpro

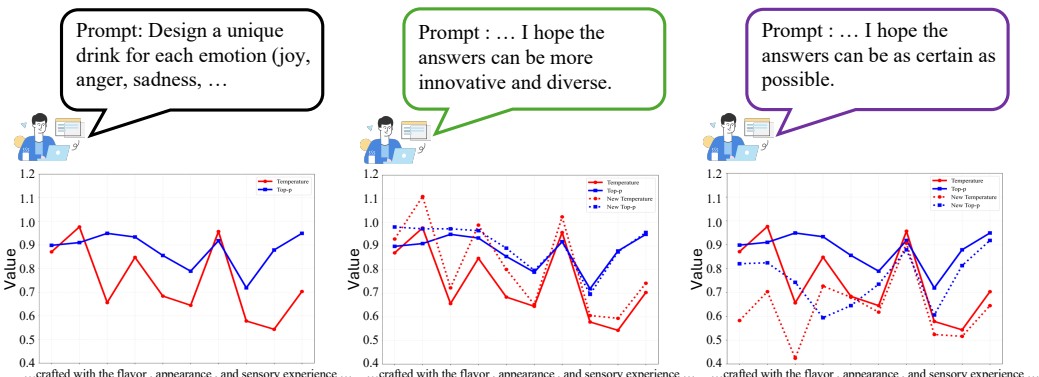

Figure 5: **An Emergent Phenomenon.** This figure shows the token-level $\hat{T}/\hat{P}$ predictions for the same prompt under three conditions, observed *without* any targeted training. **(Left) Baseline:** The model's default dynamic $\hat{T}/\hat{P}$ values. **(Middle) High-Diversity Command:** The model spontaneously elevates its $\hat{T}/\hat{P}$ predictions. **(Right) Low-Diversity Command:** The model spontaneously suppresses its $\hat{T}/\hat{P}$ predictions.

Table 4: Quantitative Impact of Diversity Commands on Predicted Decoding Parameters (N=100).

| Command | Avg. Temp. | $\triangle$ Temp. | Consistency (T) | Avg. top-p | $\triangle$ top-p | Consistency (P) |
|---|---|---|---|---|---|---|
| Baseline (No Cmd) | 0.59 | - | - | 0.84 | - | - |
| **Low Diversity** | **0.48** | $\downarrow$ **0.11** | **99%** | **0.75** | $\downarrow$ **0.09** | **99%** |
| **High Diversity** | **0.66** | $\uparrow$ **0.07** | **90%** | **0.89** | $\uparrow$ **0.05** | **96%** |

- **LiveCodeBench:** The V6 version of the dataset was used. The evaluation window for this benchmark was initiated on September 1, 2023, and included all subsequent data.

- **Others:** All the others selected datasets were processed using their full sets.

## 7  IN-DEPTH DISCUSSION ON INSTRUCTION-BASED DECODING CONTROL

The instruction-based decoding control capability is indeed an unexpected yet interesting phenomenon discovered during our experiments.

Figure 5 provides a qualitative demonstration of this capability. On the left, a creative prompt to "Design a unique drink for each emotion" elicits a dynamic but baseline set of temperature and top-p values (solid lines). In the middle panel, when we append the command, "I hope the answers can be more innovative and diverse," the model's response is immediate and visible: the predicted T and P values (dotted lines) are consistently elevated above the baseline, effectively "turning up" its own creativity. Conversely, on the right, the command "I hope the answers can be as certain as possible" causes the model to autonomously suppress its T and P predictions, "turning down" its randomness to favor more deterministic outputs. To our knowledge, this is the first demonstration of an LLM directly translating natural language intent for creativity and certainty into its internal sampling parameters on a token-by-token basis.

To verify that this is not an anecdotal result, we conducted a large-scale quantitative analysis. We prepended commands for "high" or "low" diversity to a set of 100 questions and aggregated the results, presented in Table 4. The data confirms the effect is systematic and robust. The "low diversity" command prompted a substantial drop in average temperature from 0.59 to 0.48 with remarkable **99% consistency** across all questions. The "high diversity" command triggered a similarly consistent increase in both temperature and top-p, proving that the model has learned a generalizable mapping from abstract language to its internal generation mechanics.

However, we do not yet have a conclusive understanding of this phenomenon theoretically. We will continue to advance this.

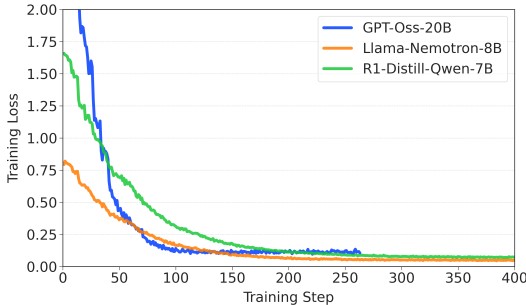

Figure 6: *AutoDeco*'s training curves on all models. Training loss curve across models. The loss converges effectively, indicating resource-friendly training of *AutoDeco*.

Table 5: Pass@16 accuracy on mathematical reasoning benchmarks. Comparing Default Sampling with the *AutoDeco* method.

| Model | Method | AIME | BRUMO25 | HMMT25 | BeyondAIME | Average |
|---|---|---|---|---|---|---|
| Llama-Nemotron-8B | Default Sampling | 77.93 | **87.98** | 61.98 | 57.36 | 71.31 |
| | *AutoDeco* (Ours) | **80.64** | 87.63 | **65.76** | **60.11** | **73.54** |
| R1-Distill-Qwen-7B | Default Sampling | 71.50 | 78.16 | 51.84 | 54.21 | 63.93 |
| | *AutoDeco* (Ours) | **74.04** | **80.08** | **58.60** | **54.88** | **66.90** |
| Qwen3-30B-Instruct | Default Sampling | 88.05 | 90.81 | **66.57** | **69.87** | 78.83 |
| | *AutoDeco* (Ours) | **88.94** | **91.15** | 66.39 | 69.55 | **79.01** |
| Qwen3-235B-Thinking | Default Sampling | 91.44 | 93.63 | **85.94** | **70.55** | 85.39 |
| | *AutoDeco* (Ours) | **92.00** | **95.81** | 85.69 | 70.00 | **85.88** |
| OpenAI-GPT-OSS-20B | Default Sampling | 91.42 | 97.10 | 86.37 | 77.32 | 88.05 |
| | *AutoDeco* (Ours) | **91.48** | **97.54** | **88.43** | **79.05** | **89.13** |

## 8 SUPPLEMENTARY DISCUSSION OF EFFICIENCY

**Training Efficiency.**    Given AutoDeco's superior decoding performance and minimal deployment overhead, a natural question arises: What is the cost of endowing a language model with this adaptive decoding capability? Remarkably, the answer is negligible. *AutoDeco* is a resource-efficient, general-purpose solution for adaptive decoding optimization. Our experiments reveal two key practical advantages:

- Label-free supervision: *AutoDeco* eliminates the need to pre-compute or invoke any external optimization modules to generate supervision signals (e.g., temperature or top-p labels) for fine-tuning.

- Data efficiency: We show the training curves of all models in Figure 6, and *AutoDeco* achieves strong performance within about only 6K training samples and 400 steps, making it effortlessly be integrated into any pre-trained LLMs.

## 9 SUPPLEMENTARY EXPERIMENTAL RESULTS

**Adaptability to Stability and Creativity.**    We selected the eight benchmarks in the main paper primarily because they are widely recognized, standard benchmarks that are frequently featured in the technical reports of leading LLMs. Furthermore, *AutoDeco* has proven effective in common open-ended generation scenarios like role-playing, enabling us to eliminate manual decoding without performance degradation. To demonstrate this, we also evaluated *AutoDeco*'s effectiveness on the RoleLLM (Wang et al., 2024a) benchmark:

- Model: Qwen3-30B-A3B-Instruct-2507.

Table 6: Pass@32 accuracy on mathematical reasoning benchmarks. Comparing Default Sampling with the *AutoDeco* method.

| Model | Method | AIME | BRUMO25 | HMMT25 | BeyondAIME | Average |
|-------|--------|------|---------|--------|------------|---------|
| Llama-Nemotron-8B | Default Sampling | 81.04 | 91.09 | 68.89 | 61.19 | 75.55 |
| | *AutoDeco* (Ours) | **84.15** | **91.57** | **71.54** | **64.72** | **78.00** |
| R1-Distill-Qwen-7B | Default Sampling | 74.24 | 80.91 | 60.65 | 59.59 | 68.85 |
| | *AutoDeco* (Ours) | **77.12** | **81.84** | **66.88** | **59.88** | **71.43** |
| Qwen3-30B-Instruct | Default Sampling | 90.22 | 93.33 | **71.03** | **74.37** | 82.23 |
| | *AutoDeco* (Ours) | **90.40** | **94.35** | 70.88 | 74.00 | **82.41** |
| Qwen3-235B-Thinking | Default Sampling | 91.80 | 95.15 | 88.96 | **73.50** | 87.35 |
| | *AutoDeco* (Ours) | **92.20** | **97.32** | **89.00** | 73.42 | **87.99** |
| OpenAI-GPT-OSS-20B | Default Sampling | 92.91 | 98.12 | 89.19 | 81.20 | 90.36 |
| | *AutoDeco* (Ours) | **93.55** | **99.14** | **91.95** | **84.36** | **92.25** |

Table 7: Pass@64 accuracy on mathematical reasoning benchmarks. Comparing Default Sampling with the *AutoDeco* method.

| Model | Method | AIME | BRUMO25 | HMMT25 | BeyondAIME | Average |
|-------|--------|------|---------|--------|------------|---------|
| Llama-Nemotron-8B | Default Sampling | 83.53 | **93.02** | **76.11** | 65.14 | 79.45 |
| | *AutoDeco* (Ours) | **87.80** | 92.41 | 74.79 | **68.87** | **80.97** |
| R1-Distill-Qwen-7B | Default Sampling | 77.75 | 81.44 | 66.25 | 65.48 | 72.73 |
| | *AutoDeco* (Ours) | **80.64** | **83.93** | **74.50** | **65.96** | **76.26** |
| Qwen3-30B-Instruct | Default Sampling | 91.15 | 95.54 | 74.31 | **78.65** | 84.91 |
| | *AutoDeco* (Ours) | **92.30** | **97.94** | **74.53** | 77.94 | **85.68** |
| Qwen3-235B-Thinking | Default Sampling | 91.67 | 96.26 | **91.24** | **77.19** | 89.09 |
| | *AutoDeco* (Ours) | **92.50** | **98.33** | 90.64 | 77.02 | **89.62** |
| OpenAI-GPT-OSS-20B | Default Sampling | 94.10 | 99.17 | 89.97 | 85.30 | 92.14 |
| | *AutoDeco* (Ours) | **95.22** | **100** | **93.97** | **88.81** | **94.50** |

Table 8: The WinRate of *AutoDeco* / fixed decoding parameters on RoleLLM.

| WinRate | $T=0, Top-p=0$ | $T=0.7, Top-p=0.8$ | $T=0.8, Top-p=0.95$ | $T=0.9, Top-p=0.95$ |
|---------|----------------|--------------------|--------------------|--------------------|
| *AutoDeco* | 54.65 / 45.35 | 54.32 / 45.68 | 53.82 / 46.18 | 52.21 / / 47.79 |

Table 9: The average predictions of *AutoDeco* in different tasks.

| | AIME | Creative Tasks |
|--|------|----------------|
| *AutoDeco* | $\hat{T}=0.61, \hat{P}=0.93$ | $\hat{T}=1.18, \hat{P}=0.88$ |

- Evaluation: We employed the advanced Claude-Sonnet-3.7 to do LLM Judge to compare generations using *AutoDeco*'s dynamic predictions against generations using several optimized fixed-parameter settings.

As demonstrated in Table 8, *AutoDeco* consistently outperforms all optimized fixed-parameter settings in these head-to-head comparisons.

We conducted an additional comparison on Qwen3-235B-A22B-Thinking-2507, between two distinct generation regimes: Mathematical Reasoning (which favors relatively high consistency/stability) and creative task (which benefits from higher variability/creativity).

- Math Results: Average results of AIME sampling.

- Creative Tasks: Average results of 20 distinct prompts. (e.g., "Please write a story about a cat" and "Help me plan a trip to Europe.")

Table 10: Pass@1 accuracy on Synthetic-100. *AutoDeco* consistently outperforms both Greedy Search and Default Sampling methods

| Model | Greedy Search | Default Sampling | *AutoDeco* (Ours) |
|---|---|---|---|
| DeepSeek-671B | 56.94 | 57.76 | **58.47** |
| DeepSeek-685B (with MTP) | 57.12 | 57.89 | **58.40** |

The average predictions shown in Table 9 demonstrates the ability of *AutoDeco* to adapt to different task requirements.

**Pass@k Performance.** With encouraging results, we find that the absolute improvements delivered by *AutoDeco* at higher $k$-values are consistent with, and at times even slightly greater than, those observed at pass@1.

It is important to note that for any given model, pass@$k$ accuracy is inherently much higher than pass@1 accuracy. Consequently, it is obvious that securing absolute performance gains becomes substantially more difficult, and a similar absolute improvement at pass@64 translates to a much larger relative error reduction, compared to pass@1. For example, on the OpenAI-GPT-OSS-20B model, we observe that the performance gains from *AutoDeco* are consistent across different $k$ values in pass@$k$ evaluations. More importantly, this consistent absolute gain translates to a significantly larger impact in higher-accuracy (when k is large) scenarios. The relative error reduction dramatically increases from **3.2%** at pass@1 to **30%** at pass@64. This demonstrates that as the task becomes easier for the baseline model (i.e., the error rate decreases at high $k$), the performance gains from our method become even more significant.

**DeepSeek-V3.1-Terminus.** The performance gains achieved on production-level models are our main focus. We have also conducted performance evaluations on industrial-grade SOTA large models such as DeepSeek-V3.1-Terminus-671B (DeepSeek-AI, 2024).

Due to the deployment and sampling pressure of the ultra-large thinking model, we combined all the evaluated benchmarks on an average basis and created a tiny synthetic benchmark consisting of 100 evaluation questions (named Synthetic-100) for evaluating *AutoDeco*.

Our method is fully compatible and integrates seamlessly with speculative decoding mechanisms. Taking MTP as an example, it is fundamentally still of the form of autoregression. Sampling parameters such as Temperature and Top-p are widely and continuously employed within it to control the diversity and quality of the final accepted tokens.

To empirically confirm this seamless integration, we conducted a MTP experiment on the advanced LLM DeepSeek-V3.1-Terminus and demonstrate the superiority of *AutoDeco* in Table 10. It proves that using *AutoDeco* will not disrupt the MTP process of the model. Critically, we find that the sampling parameters temperature and top-p are still of vital importance and do not diminish as the size and capability of the model increase. *AutoDeco* can help users achieve a descent performance with the least effort.

## 10 DECLARATION OF LLM USAGE

The LLM is used only for writing, editing, or formatting purposes and does not impact the core methodology, scientific rigorousness, or originality of the research.

Table 11: Pass@1 accuracy on mathematical reasoning benchmarks. *AutoDeco* consistently outperforms both Greedy Search and Default Sampling methods across various models.

| Model | Method | AIME | BRUMO25 | HMMT25 | BeyondAIME | Average |
|---|---|---|---|---|---|---|
| Llama-Nemotron-8B | Greedy Search | 55.00 | 60.00 | 26.67 | **37.00** | 44.67 |
| | Default Sampling | 51.24±1.31 | 60.68±1.21 | 30.89±1.29 | 32.65±0.51 | 43.86 |
| | *AutoDeco* (Ours) | **56.07±1.12** | **63.23±1.04** | **34.37±1.33** | 34.51±0.54 | **47.05** |
| R1-Distill-Qwen-7B | Greedy Search | 40.00 | 43.33 | 20.00 | 24.00 | 31.83 |
| | Default Sampling | 43.62±1.06 | 51.38±1.08 | 22.50±0.88 | 24.87±0.72 | 35.59 |
| | *AutoDeco* (Ours) | **47.57±1.20** | **53.93±1.31** | **24.46±0.76** | **26.98±0.75** | **38.24** |
| Qwen3-30B-Instruct | Greedy Search | 65.00 | 70.00 | 36.67 | 45.00 | 54.17 |
| | Default Sampling | 66.95±1.44 | 70.99±1.11 | **44.69±0.81** | 47.02±0.33 | 57.41 |
| | *AutoDeco* (Ours) | **68.22±0.85** | **71.36±0.83** | 43.15±1.10 | **47.15±0.20** | **57.47** |
| Qwen3-235B-Thinking | Greedy Search | 80.00 | 83.33 | 63.33 | **53.00** | 69.92 |
| | Default Sampling | 80.76±0.43 | 82.40±0.69 | 63.41±1.20 | 50.05±0.56 | 69.16 |
| | *AutoDeco* (Ours) | **82.79±0.72** | **84.14±1.27** | **66.07±1.10** | 51.88±0.62 | **71.22** |
| OpenAI-GPT-OSS-20B | Greedy Search | 56.67 | 73.33 | **56.67** | 37.00 | 55.92 |
| | Default Sampling | 69.65±1.33 | 75.50±1.11 | 50.91±2.76 | 47.11±0.23 | 60.79 |
| | *AutoDeco* (Ours) | **72.72±1.22** | **75.21±0.83** | 53.00±1.02 | **47.32±0.37** | **62.06** |
| OpenAI-GPT-OSS-120B | Greedy Search | 78.50 | 66.67 | **66.67** | 50.00 | 65.46 |
| | Default Sampling | 78.33±0.72 | 78.23±0.91 | 63.17±1.80 | **53.45±0.79** | 68.30 |
| | *AutoDeco* (Ours) | **78.65±0.46** | **78.55±1.35** | 62.90±1.19 | 53.24±0.63 | **68.34** |

Table 12: Pass@1 accuracy on general-domain benchmarks. *AutoDeco* shows exciting generalization performance across General QA, Code Generation, and Instruction Following tasks.

| Model | Method | GPQA-Diamond | MMLU-Pro | LiveCodeBenchV6 | IFEval | Average |
|---|---|---|---|---|---|---|
| Llama-Nemotron-8B | Greedy Search | **51.01** | 52.00 | 19.17 | **71.53** | 48.43 |
| | Default Sampling | 44.93 | 54.00 | 21.22 | 65.25 | 46.35 |
| | *AutoDeco* (Ours) | 50.52 | **55.64** | **21.68** | 71.02 | **49.72** |
| R1-Distill-Qwen-7B | Greedy Search | 37.87 | 47.20 | 14.98 | 32.90 | 33.24 |
| | Default Sampling | 47.41 | 47.65 | 15.40 | 32.35 | 35.70 |
| | *AutoDeco* (Ours) | **48.91** | **50.75** | **15.46** | **33.90** | **37.26** |
| Qwen3-30B-A3B-Instruct-2507 | Greedy Search | 65.86 | 78.00 | 47.75 | **83.73** | 68.84 |
| | Default Sampling | 69.82 | 76.25 | 48.52 | 81.52 | 69.03 |
| | *AutoDeco* (Ours) | **69.96** | **78.38** | **49.80** | 82.81 | **70.24** |
| Qwen3-235B-Thinking | Greedy Search | 77.78 | **81.00** | 77.25 | 31.98 | 67.00 |
| | Default Sampling | 80.81 | 80.33 | 77.47 | 31.61 | 67.56 |
| | *AutoDeco* (Ours) | **81.13** | 79.20 | **78.10** | **32.90** | **67.83** |
| OpenAI-GPT-OSS-20B | Greedy Search | 59.60 | 67.00 | 69.69 | 29.94 | 56.56 |
| | Default Sampling | 65.67 | 68.00 | 70.15 | 30.68 | 58.63 |
| | *AutoDeco* (Ours) | **66.48** | **69.12** | **71.25** | **30.84** | **59.42** |
| OpenAI-GPT-OSS-120B | Greedy Search | **71.21** | 75.00 | 73.93 | 32.72 | 63.22 |
| | Default Sampling | 70.20 | 76.00 | 74.19 | 32.53 | 63.23 |
| | *AutoDeco* (Ours) | 70.24 | **76.50** | **74.30** | **32.90** | **63.49** |

