# OpenReview forum: "THE END OF MANUAL DECODING: TOWARDS TRULY END-TO-END LANGUAGE MODELS"
_ICLR.cc/2026/Conference — ICLR 2026 Poster_

### Official Review · Reviewer_8WUM · 2025-10-21

**Soundness:** 3
**Presentation:** 3
**Contribution:** 3
**Rating:** 4
**Confidence:** 4

**Summary:**

The paper introduces AutoDeco, a framework that aims to make language model decoding fully end-to-end by enabling models to predict their own decoding hyperparameters (temperature and top-p) dynamically at each generation step. The authors augment standard transformers with lightweight MLP heads that output token-specific decoding parameters, trained using pseudo-labels derived from optimization on ground-truth tokens. The approach claims to remove the need for manual or oracle-style hyperparameter tuning while maintaining negligible computational overhead.

**Strengths:**

1. Novel framing of decoding as a learnable, differentiable process. Treating temperature and top-p as learnable functions of context rather than fixed hyperparameters is an appealing conceptual shift that aligns with recent trends toward self-regulating and adaptive LLM inference.
2. Simple yet effective architecture. The addition of two MLP heads is computationally cheap and can be easily integrated into existing transformer architectures, which improves the practicality and reproducibility of the approach.
3. The observation that the model can interpret natural-language modifiers like “low randomness” to adjust temperature/top-p values is intriguing and opens a new line of research in interpretable controllability.

**Weaknesses:**

1. My concern is that it is not clear to me how they obtain labels for training prediction of temperature and top_p value per token. It is unclear how the argmax over continuous T > 0 is solved, what constraints or search ranges are used, and how noise in logits affects these derived labels. It is better to provide a simple example to explain this process.
2. The experiments only compare against Greedy and Default Sampling (and an oracle). Missing are modern decoding methods such as Contrastive Search [1], Contrastive Decoding [2].

[1] https://arxiv.org/pdf/2210.14140
[2] https://arxiv.org/abs/2309.09117

**Questions:**

Please see the weaknesses

---

> ### Author Response · Authors · 2025-11-20
>
> We sincerely thank you for your time and suggestions.
>
> ### **For Weakness 1**
>
> We have substantially revised and expanded the **"Training Strategy"** of Section 2 in the manuscript to include a fully end-to-end training and a clearer step-by-step data flow.
>
> In fact, this is consistent with the optimization goal of traditional SFT of LLM, but only the parameters of the AutoDeco Heads are updated.
>
> For example, for every token $ y\_t^* $ in the ground-truth sequence (where t is the t-th token position), our objective is to find the pair of hyperparameters $ (\hat{T}^\*, \hat{P}^\*) $  that maximizes the probability of sampling that exact token $ y\_t^* $ from the base model's distribution at that step. This is the definition of our argmax objective:$(\hat{T}^\*, \hat{P}^\*) = \arg\mathop{\max}\_{\hat{T}>0, \hat{P} \in (0, 1]} P(y_t^\* | \text{context}, T, P)$.
>
> ### **For Weakness 2**
>
> First, we would like to clarify why Contrastive Search (CS) and Contrastive Decoding (CD) were not included as our baselines. These methods are primarily designed to mitigate the 'degeneration' problem, which has become significantly less prevalent in modern instruction-tuned models. Furthermore, active maintenance for these decoding algorithms has ceased in the latest versions of major libraries such as Hugging Face Transformers and vLLM.
>
> We also conducted quantitative experiments to validate our claim. As shown in the table below, on the HMMT25 dataset, CS and CD (using recommended parameters) did not yield effective performance improvements compared to our established sampling baselines, and they were outperformed by AutoDeco’s dynamic parameter prediction.
>
> Table 1: Pass@1 accuracy on the HMMT 25 with Contrastive Search. ($\alpha=0.6$, top-k=6)
>
> | **Model** | **Greedy Search** | **Default Sampling** | **CS** | **Ours** |
> | --- | --- | --- | --- | --- |
> | Llama-Nemotron-8B | 26.67 | 29.82 | 26.67 | **33.98** |
> | R1-Distill-Qwen-7B | 16.67 | 22.32 | 16.67 | **24.14** |
> | Qwen3-30B-Instruct | 36.67 | **43.88** | 40 | 43.73 |
> | Qwen3-235B-Thinking | 63.33 | 61.95 | 63.33 | **64.17** |
> | OpenAI-GPT-OSS-20B | **46.67** | 44.24 | 36.67 | 46.2 |
>
> Table 2: Qwen3-235B-A22B-Thinking-2507 pass@1 accuracy on the HMMT 25 with Contrastive Decoding. (amateur model: Qwen3-30B-A3B-Thinking-2507, $\alpha=0.1, \beta=0.5$)
>
> |  | **Greedy Search** | **Default Sampling** | **CD** | **Ours** |
> | --- | --- | --- | --- | --- |
> | HMMT 25 | 63.33 | 61.95 | 63.33 | **64.17** |
>
> ---
>
> We hope our detailed responses have thoroughly addressed your concerns, and we are happy to resolve any questions you may have. In addition to these points, we have also made significant revisions that we believe, 'fundamentally strengthen' the manuscript (please refer to our general comments "Initial Updates to Manuscript"). Given these clarifications and the substantial improvements to the paper, we believe the work merits a rating higher than its current one and respectfully ask that you consider this in your final evaluation.

---

> > ### Comment · Reviewer_8WUM · 2025-11-22
> >
> > Thanks for the response and new experiment results.
> > My concerns have been addressed. I updated my scores accordingly.

---

### Official Review · Reviewer_tpjS · 2025-10-27

**Soundness:** 3
**Presentation:** 1
**Contribution:** 3
**Rating:** 6
**Confidence:** 3

**Summary:**

The paper introduces a method to learn the sampling hyperparameters (temp, top-p) end to end via finetuning, avoiding manual task-specific tuning. They also identify and give ways to solve challenges that arise when trying to do this naively. In the end, they find they can basically match task-specific manual tuning (the oracle upper bound).

**Strengths:**

- The idea is interesting and to my knowledge novel.
- The problem space is richer than meets the eye (eg. how do you get training data for supervised training is surprisingly nontrivial).
- The results are fairly convincing (Fig2).
- They show this doesn't hurt performance (Fig3).

**Weaknesses:**

- In practice, nobody does pure autoregression in real world LLM usage at production-scale. Everyone uses speculative decoding of some sort, and it's not clear to me whether this sampling scheme permits that or breaks it. I would want to see an explanation or formal argument/construction for how speculative decoding would work when the target model has an AutoDeco head to be convinced this would not break speculation. Because if it does, then in practice it will never be used, which would be a major limitation. I think it can be made to work, though, it just needs maybe some explanation of how it would all come together.
- The idea and science is good, but writing is mediocre in my opinion. The contributions section repeats the second contribution twice, omitting I imagine the third contribution (presumably the "emergent instruction-following"), the plots are not consistently formatted well (axes in Fig4 impossible to read), the name "AutoDeco" is neither catchy nor informative in my opinion, etc.
- The "emergent ability" is not as surprising as the authors seem to think it is. You're training on a base model that conditions on language by construction, so I would expect this behavior as the language conditioning affects the model representations fed into the AutoDeco head. Maybe de-emphasize this, or figure out an alternative framing? But this is minor.

**Questions:**

See weaknesses.

---

> ### Author Response · Authors · 2025-11-20
>
> We are deeply grateful for your highly insightful and constructive review. Thank you for introducing the critical context of Speculative Decoding into our discussion.
>
> ### **For weakness 1**
>
> Our method is **fully compatible** and integrates seamlessly with speculative decoding mechanisms. Taking MTP as an example, it is fundamentally still of the form of autoregression. Sampling parameters such as Temperature and Top-p are widely and continuously employed within it to control the diversity and quality of the final accepted tokens.
>
> To empirically confirm this seamless integration, we conducted a multi-token prediction experiment on the advanced LLM **DeepSeek-V3.1-Terminus** in Table 1.
>
> Table 1: MTP Decoding with AutoDeco on Synthetic-100. Our method can seamlessly integrate the advanced speculative decoding process (MTP).
>
> |  | **Greedy Search** | **Default Sampling** | **AutoDeco (Ours)** |
> | --- | --- | --- | --- |
> | DeepSeek-V3.1 - Terminus | 56.94 | 57.76 | **58.47** |
> | DeepSeek-V3.1 - Terminus-MTP | 57.12 | 57.89 | **58.40** |
>
> It shows that our method does not require any specific adaptation and will not break the process of speculative decoding. We have already supported MTP decoding with AutoDeco in the vLLM, and are looking forward to sharing it with the community.
>
> ### **For weakness 2**
>
> We sincerely apologize for the current presentation errors. We have thoroughly revised the manuscript to address the presentation concerns.
>
> ### **For weakness 3**
>
> We appreciate the reviewer's recognition and kind suggestion. Our initial intention was to showcase AutoDeco's primary contribution: **truly end-to-end generation** by learning to predict optimal decoding hyperparameters for every generation step, thus eliminating the manual, non-differentiable tuning process.
>
> To ensure our claims are appropriately scoped and to avoid any perception of overclaiming our contribution, we have revised the manuscript to: 1) reduce the emphasis on the "instruction-based decoding control" capability in the main paper, and 2) move the detailed discussion of this phenomenon to Appendix 7.
>
> ---
>
> Thank you sincerely for your kind evaluation and constructive suggestions. Based on our communication, we have added a comprehensive discussion on our method with Speculative Decoding in Section 3.2 pragraph **Compatibility with Advanced Decoding Algorithms**, and enhanced the presentation. We remain readily available to discuss any further questions or concerns with you.
>
> In addition to these points, we have also made significant revisions that we believe, 'fundamentally strengthen' the manuscript (please refer to our general comments "Initial Updates to Manuscript"). We sincerely hope you will reconsider your current evaluation and recognize the merits of the revised work.

---

### Official Review · Reviewer_zLkm · 2025-11-01

**Soundness:** 3
**Presentation:** 3
**Contribution:** 3
**Rating:** 6
**Confidence:** 2

**Summary:**

This work addresses an important practical issue in deploying large language models (LLMs): the need to manually tune decoding hyperparameters, which breaks the “end-to-end” workflow. To mitigate this, the authors propose AutoDeco, a method that augments standard transformer architectures with lightweight prediction heads that output context-dependent temperature and top-p values alongside next-token logits during decoding. Experiments across multiple benchmarks demonstrate that the approach can yield improved generation quality and adaptability.

**Strengths:**

1. The problem is timely and relevant. Removing manual tuning of decoding hyperparameters can significantly improve the practicality and usability of LLM-based systems.

2. The pseudo-label generation strategy for supervision is clever and helps circumvent the lack of direct ground-truth hyperparameter labels.

**Weaknesses:**

1. A key concern is that the model is trained to further increase the likelihood of the reference text. Since both pre-training and downstream fine-tuning typically already optimize for the likelihood of the ground-truth sequence, this additional adjustment may risk overfitting or reduce robustness in more open-ended generation settings.

2. While dynamic prediction of decoding hyperparameters is appealing, different applications may require different behavior. For example, customer support systems typically favor stability and consistency, whereas brainstorming or creative writing tasks may benefit from higher variability. It is unclear whether a single learned mechanism can generalize effectively across such diverse usage scenarios without explicit control, and it may limit adaptability when user preferences change.

**Questions:**

1. Can this framework be extended to other decoding parameters, such as top-k or repetition penalty?

2. In practical deployments, how should developers express or configure user-level preferences (e.g., consistency vs. creativity)? More discussion or empirical analysis would clarify how the method adapts to varying application requirements, especially in relation to the concern raised above.

---

> ### Author Response · Authors · 2025-11-20
>
> We sincerely thank Reviewer zLkm for recognizing the timeliness and relevance of our work, as well as for the kind comments regarding our strategy.
>
> ### **For weakness 1**
>
> This is also the key point we are focusing on, and we believe the experimental results have partially addressed this concern.
>
> Despite the restricted training domain, AutoDeco consistently outperforms all baselines across **out-of-domain** tasks. This suggests **satisfactory generalization and robustness.**
>
> Furthermore, in our own industrial practice, AutoDeco has proven effective in common open-ended generation scenarios like role-playing, enabling us to eliminate manual decoding without performance degradation. To demonstrate this, we also evaluated AutoDeco's effectiveness on the RoleLLM [1] benchmark:
>
> - Model: Qwen3-30B-A3B-Instruct-2507.
> - Evaluation: We employed the advanced Claude-Sonnet-3.7 to do LLM Judge to compare generations using AutoDeco’s dynamic predictions against generations using several optimized fixed-parameter settings.
>
>
> Table 1: The WinRate of AutoDeco / fixed decoding parameters. The left result is the rate of AutoDeco, and the right result is of baseline.
> | WinRate  | T=0.0, Top-p=0.0 | T=0.7, Top-p=0.8 | T=0.8, Top-p=0.95 | T=0.9, Top-p=0.95 |
> | --- | --- | --- | --- | --- |
> | **AutoDeco** | 54.65 / 45.35 | 54.32 / 45.68 | 53.82 / 46.18 | 52.21 / 47.79 |
>
> Obviously, as demonstrated in Table 1, AutoDeco consistently outperforms all optimized fixed-parameter settings in these head-to-head comparisons.
>
> ### **For weakness 2**
>
> The model has the ability to adapt to different task requirements. To confirm this, we conducted an additional comparison on Qwen3-235B-A22B-Thinking-2507, between two distinct generation regimes: **Mathematical Reasoning** (which favors relatively high **consistency/stability**) and **creative task** (which benefits from higher **variability/creativity**).
>
> - Math Results: Average results of AIME sampling.
> - Creative Tasks: Average results of 20 distinct prompts. (e.g., "Please write a story about a cat" and "Help me plan a trip to Europe.")
>
> Table 2: The average predictions of AutoDeco in different tasks.
>
> |  | **AIME** | **Creative Tasks** |
> | --- | --- | --- |
> | **AutoDeco** | $\hat{T}=0.61, \hat{P}=0.93$ | $\hat{T}=1.18, \hat{P}=0.88$ |
>
> We hope that the model's adaptability to different tasks will further address your concerns.
>
> ### **For question 1**
>
> With our end-to-end training approach, top-k and repetition penalty applications are all differentiable throughout the entire process, and thus are completely scalable.
>
> ### **For question 2**
>
> The primary goal of AutoDeco is to ensure that the user or developer **does not need to manually tune** decoding parameters to achieve a **decent, generalized performance** across varied tasks.
>
> Meeting diverse user needs is an interesting product choice, yet both paths are technically easy to implement.
>
> As developers, if the goal is to relieve users from the burden of tuning decoding hyperparameters, you can simply remove parameters such as temperature and top_p from the API. Conversely, if you wish to retain user agency in choosing between creative and deterministic outputs, you can utilize AutoDeco’s predictions as the default setting, allowing user-defined hyperparameters to either override these defaults or rescale the target distribution.
>
> ---
>
> Thank you for your contribution, which has pushed our paper to a more robust state. We hope that our detailed comparative experiments can address your concerns about the adaptability of AutoDeco.  We have added these discussions to the revision Section 3.2.1 (paragraph on "out-of-domain performance"), and we are very pleased to further discuss with you the issues and concerns you may have.
>
> In addition to these points, we have also made significant revisions that we believe, 'fundamentally strengthen' the manuscript (please refer to our general comments "Initial Updates to Manuscript"). Given these comprehensive clarifications and the depth of the improvements made to the paper, we sincerely hope you will reconsider your current evaluation and recognize the merits of the revised work.
>
> [1] Wang, Noah, et al. "Rolellm: Benchmarking, eliciting, and enhancing role-playing abilities of large language models." *Findings of the Association for Computational Linguistics: ACL 2024*. 2024.

---

### Official Review · Reviewer_XeNy · 2025-11-02

**Soundness:** 2
**Presentation:** 2
**Contribution:** 2
**Rating:** 4
**Confidence:** 2

**Summary:**

The paper introduces AutoDeco, a novel architecture designed to make large language models (LLMs) truly end-to-end by eliminating the need for manual decoding hyperparameter tuning (e.g., temperature, top-p). Traditional decoding strategies rely on static, hand-tuned parameters that must be manually adjusted for different tasks or even different parts of a generation. AutoDeco addresses this by augmenting the transformer with lightweight “decoding heads” that dynamically predict temperature and top-p values at every generation step.

**Strengths:**

Novelty and Conceptual Contribution: The paper identifies and addresses a fundamental yet overlooked bottleneck in LLM deployment, the manual, non-differentiable decoding process.

AutoDeco reframes decoding as a learnable and parametric component within the model itself, offering a principled step toward fully end-to-end generation.

**Weaknesses:**

1. While the emergent instruction-following behavior is a highlight, the explanation for why this arises is mostly empirical. A deeper analysis (e.g., probing whether linguistic cues correlate with latent space adjustments) would strengthen this claim.


2. Most benchmarks are reasoning or QA-oriented. It would be valuable to test AutoDeco on creative writing, dialogue, or long-form generation, where decoding choices play a larger role. Human evaluation or qualitative examples of improved text quality would strengthen the practical impact.

**Questions:**

Please refer to the weakness part.

---

> ### Author Response · Authors · 2025-11-20
>
> We sincerely thank Reviewer XeNy for the thoughtful and constructive feedback on our submission.
>
> ### **For weakness 1**
>
> We appreciate the reviewer's insight. Our initial intention was to showcase AutoDeco's primary contribution: **truly end-to-end generation** by learning to predict optimal decoding hyperparameters for every generation step, thus eliminating the manual, non-differentiable tuning process.
>
> The " instruction-based decoding control" capability was indeed an unexpected yet interesting phenomenon discovered during our experiments. We were quite enthusiastic about this finding and eager to share it. However, we must admit that we currently lack a formal explanation beyond these empirical observations (an area we are still working on in progress).
>
> Therefore, to ensure our claims are appropriately scoped and to avoid any perception of overclaiming our contribution, we have revised the manuscript to: 1) reduce the emphasis on the "instruction-based decoding control" capability in the main paper, and 2) move the detailed discussion of this phenomenon to Appendix 7.
>
> ### **For weakness 2**
>
> We selected the eight benchmarks in the main paper primarily because they are widely recognized, standard benchmarks that are frequently featured in the technical reports of leading LLMs.
> Furthermore, in our own industrial practice, AutoDeco has proven effective in common open-ended generation scenarios like role-playing, enabling us to eliminate manual decoding without performance degradation. To demonstrate this, we also evaluated AutoDeco's effectiveness on the RoleLLM [1] benchmark:
>
> - Model: Qwen3-30B-A3B-Instruct-2507.
> - Evaluation: We employed the advanced Claude-Sonnet-3.7 to do LLM Judge to compare generations using AutoDeco’s dynamic predictions against generations using several optimized fixed-parameter settings.
>
> Table 1: The WinRate of AutoDeco / fixed decoding parameters. The left result is the rate of AutoDeco, and the right result is of baseline.
>
> | WinRate  | T=0.0, Top-p=0.0 | T=0.7, Top-p=0.8 | T=0.8, Top-p=0.95 | T=0.9, Top-p=0.95 |
> | --- | --- | --- | --- | --- |
> | **AutoDeco** | 54.65 / 45.35 | 54.32 / 45.68 | 53.82 / 46.18 | 52.21 / 47.79 |
>
> Obviously, as demonstrated in Table 1, AutoDeco consistently outperforms all optimized fixed-parameter settings in these head-to-head comparisons.
>
> ---
>
> We are, of course, happy to discuss any further questions you may have. We hope our detailed responses have thoroughly addressed your concerns. In addition to these points, we have also made significant revisions that we believe, 'fundamentally strengthen' the manuscript (please refer to our general comments "Initial Updates to Manuscript"). Given these clarifications and the substantial improvements to the paper, we believe the work merits a rating higher than its current one and respectfully ask that you consider this in your final evaluation.
>
> [1] Wang, Noah, et al. "Rolellm: Benchmarking, eliciting, and enhancing role-playing abilities of large language models." *Findings of the Association for Computational Linguistics: ACL 2024*. 2024.

---

### Author Response · Authors · 2025-11-12
**Initial Updates to Manuscript**

Dear Reviewers and ACs,

Thank you for your time and valuable feedback.

We have uploaded a significantly revised version of our paper, reflecting major progress made since the initial submission. We are confident these updates fundamentally strengthen our work and address key concerns.

Here are the highlights of the revision:

A Fundamental Leap to True End-to-End Training: This is the most significant improvement. We have evolved our method from a cascaded pipeline (which relied on generating pseudo-labels) to a true end-to-end training paradigm. As detailed in the revised Section 2.1, gradients from the standard cross-entropy loss are now backpropagated to directly and simultaneously update both the temperature and top-p heads. This methodological leap is crucial: it creates a more elegant and robust training process, completely removing the need for intermediate, potentially noisy supervision signals. This new paradigm fully resolves the core question of label acquisition, as raised by Reviewer 8WUM (Weakness 1).

Expanded Evaluation on State-of-the-Art Large Models: To prove the scalability and practical value of AutoDeco, we have extended our validation to much larger, production-scale models. The revised manuscript now includes comprehensive results for Qwen3-235B-A22B-Thinking-2507 and DeepSeek-V3.1-Terminus-671B (Tables 1, 2, 8). These new experiments confirm that AutoDeco's effectiveness holds strong even on today's most powerful architectures.

Comprehensive Robustness Analysis with pass@k: To provide a more rigorous assessment beyond pass@1, we introduced an extensive pass@k analysis (Appendix 8, Tables 5, 6, 7). The results are highly encouraging: we find that the absolute performance improvements delivered by AutoDeco are not only sustained but often become even greater at higher k-values. This demonstrates the robustness of our method, proving it enhances the overall quality of the entire solution space, not just the single best answer.

Given these fundamental improvements, we believe the paper is in a much stronger position. We respectfully ask you to re-evaluate our work based on this new, more powerful version.

We will now proceed to address your individual comments and look forward to a constructive discussion.

Best regards,

The Authors

---

### Author Response · Authors · 2025-11-20
**Update of the Manuscript According to the Rebuttal Discussion**

Dear Reviewers and ACs,

We sincerely thank you for your time and insightful feedback, which have been invaluable in improving our manuscript. We have carefully considered all comments and have provided detailed responses to each reviewer individually.

Below, we summarize our main revisions addressing the common points:

**On the Adaptability of AutoDeco:** We selected the eight benchmarks in the main paper primarily because they are widely recognized, standard benchmarks that are frequently featured in the technical reports of leading LLMs. Furthermore, in our own industrial practice, AutoDeco has proven effective in common open-ended generation scenarios like role-playing, enabling us to eliminate manual decoding without performance degradation.

We have conducted further evaluations to assess the adaptability of our method. We benchmarked AutoDeco on new **Creative Tasks** and **Role-play** scenarios. The results, which demonstrate the **strong out-of-domain generalization capabilities** of our approach, have now been incorporated into the main paper in **Sec 3.2.1** (paragraph on "out-of-domain performance").

**On Instruction-Based Decoding Control:** We thank Reviewer **XeNy** and **tpjS** for their valuable feedback on this topic. It was indeed an unexpected yet interesting phenomenon discovered during our experiments. While we were enthusiastic about this preliminary finding, we must candidly acknowledge that we currently lack a comprehensive explanation beyond these empirical observations, and this remains an area of ongoing work.

Therefore, to ensure our claims are appropriately scoped and to avoid overclaiming our contribution, we have revised the manuscript as follows:

- We have toned down the emphasis on the "instruction-based decoding control" capability in the main paper.
- The detailed discussion of this phenomenon has been moved to **Appendix 7**.

The revised manuscript has been uploaded. We believe these revisions substantially strengthen the paper and more accurately reflect our core contributions, and we respectfully request your re-evaluation. Thanks again for the constructive suggestions to improve our work, and look forward to your further feedback.

Best regards,

The Authors

---

### Author Response · Authors · 2025-11-27

Dear Reviewers and ACs,

We sincerely appreciate the reviewers’ thoughtful feedback and thank the AC for the efforts behind the scenes.

We would like to respectfully note that most of the current scores were given prior to our rebuttal and the substantially updated manuscript. We sincerely request that you re-evaluate the merit and scoring of our work in light of our detailed responses:
- The newly added results (on open-ended dialogue) demonstrate the broad **adaptability and generality** of our method across diverse tasks.
- Our method can be **seamlessly integrated with speculative decoding**, and we have already provided empirical validation on advanced MTP strategies.
- The **breakthrough fully end-to-end training strategy** is presented in the revised Section 2.

The earlier concerns (e.g., adaptability, speculative decoding, instruction-based decoding control) are now **cleanly resolved in the new version**. We believe these updates significantly strengthen the work and more accurately reflect its contribution and technical value.

We truly appreciate your time and kind evaluation, and believe the clarified results and methodology will support a positive re-evaluation.

Best regards,

The Authors

---

### Author Response · Authors · 2025-12-02
**Summary of Rebuttal & Updates for the New AC**

Dear Area Chair,

We **sincerely appreciate your effort in handling this submission, especially given the extra workload caused by the recent re-assignment.** To assist your efficient decision-making, we provide a concise summary of our rebuttal progress:

**1. Positive Outcome from Active Engagement**

Crucially, **Reviewer 8WUM had acknowledged our responses and updated manuscript by raising their score to 6 (Confidence 4) 5 days prior (Nov. 22)** to the platform bug.

**2. Cleanly Resolved Concerns for Remaining Reviewers**

Although other reviewers (Confidence 2-3) have not yet engaged, **we strongly believe their concerns are fully resolved** by our new experiments and revision, and we anticipate a positive re-evaluation upon their review:

- **Adaptability (Reviewer XeNy & zLkm):** We originally focused on benchmarks with standard answers for rigor. To address their concern, we added **Role-Playing and Stability vs. Creativity experiments** (Appendix 9 of the revised manuscript), which clearly prove that **AutoDeco works perfectly on open-ended tasks** as well.
- **Personal Control (Reviewer zLkm):** We clarified that supporting user-defined preferences is a product choice, not a technical blocker. We offered a detailed implementation path that is technically effortless, which **definitely dispels this concern**.
- **Speculative Decoding (Reviewer tpjS):** We are excited to report that we have successfully verified AutoDeco on the latest **DeepSeek-V3.1 with MTP**. This proves seamless compatibility with advanced speculative decoding—a result we believe Reviewer tpjS will be **delighted to see**.

**3. Major Technical Breakthrough in Revision**

Beyond clarifications, we have significantly upgraded the method since the initial reviews:

- **True End-to-End Training:** We introduced a breakthrough fully end-to-end training strategy (Revised Section 2), eliminating intermediate steps.
- **SOTA Scalability:** We validated performance on production-scale models, including **Qwen3-235B and DeepSeek-685B**.

We are confident that AutoDeco represents a fundamental innovation (as every reviewer has acknowledged the novelty of our approach and the effectiveness of our framework) in decoding architecture and is poised to become a standard component for future SOTA models.

We once again thank you for your time and fair evaluation during this critical time.

Best regards,

The Authors

---

### Meta-Review · Area_Chair_TZQt · 2026-01-08

**Summary:**

**Strengths**

1. Novel conceptual framing: Recasts decoding as a learnable, differentiable, and parametric component rather than a manually tuned post-hoc procedure, addressing a real deployment bottleneck (XeNy, 8WUM).

2. Practical motivation: Removing manual decoding hyperparameter tuning improves the usability and practicality of LLM systems across applications (zLkm).

3. Lightweight and integrable design: The approach adds only simple MLP prediction heads, making it easy to incorporate into existing models without high computational overhead (8WUM).

4. Pseudo-labeling strategy for training: Clever use of pseudo-supervision circumvents the lack of ground-truth temperature/top-p labels (zLkm).

5. Empirical evidence that performance is preserved: Results suggest that dynamic decoding does not degrade core model accuracy (tpjS).

6. Natural-language controllability: The observation that the model can react to modifiers like “low randomness” reveals an intriguing direction toward interpretable controllability (XeNy, 8WUM).

**Weaknesses**:

1. Limited explanation of emergent behavior: Instruction-following effects are described empirically with little mechanistic analysis, leaving open how language cues influence the latent decoding dynamics (XeNy).

2. Narrow evaluation scope: Benchmarks focus on QA/reasoning tasks rather than settings where decoding matters most (e.g., creative writing, dialogue, long-form generation, human evaluations) (XeNy).

3. Potential overfitting to reference likelihood: Training to further increase reference likelihood may reduce robustness or diversity in open-ended generation scenarios (zLkm).

4. Lack of adaptability across usage preferences: Different real-world applications favor different decoding behaviors; it is unclear how a single learned mechanism adapts without explicit user control (zLkm).

5. Unclear compatibility with speculative decoding: Practical deployment often depends on speculative decoding; it is unclear whether AutoDeco is compatible, limiting real-world utility (tpjS).

6. Presentation and polish issues: Writing clarity, contribution statement, and figure formatting inconsistencies detract from readability and accessibility (tpjS).

7. Label construction ambiguity: The process for obtaining continuous temperature/top-p pseudo-labels lacks detail regarding search, constraints, and noise sensitivity (8WUM).

8. Missing comparisons to modern decoding baselines: Contrastive decoding and other contemporary decoding methods are not evaluated, limiting claims of improvement (8WUM).

**Reviewer Concerns:**

Most concerns have been adequately addressed during the rebuttal. The remaining point, namely the limited mechanistic explanation of the emergent instruction-following behavior, suggest that additional analysis would clarify how language cues influence latent decoding dynamics, but this does not constitute a critical limitation for acceptance.

**Reviewer Scores:**

- Reviewer XeNy: 4 -> 6
- Reviewer zLkm: 6 - > 6
- Reviewer tpjS: 6 -> 6
- Reviewer 8WUM: 4 -> 6

---

### Decision · Program_Chairs · 2026-01-26

Accept (Poster)